# Terahertz Spectroscopic Identification of Roast Degree and Variety of Coffee Beans

**DOI:** 10.3390/foods13030389

**Published:** 2024-01-24

**Authors:** Luelue Huang, Miaoling Liu, Bin Li, Bimal Chitrakar, Xu Duan

**Affiliations:** 1School of Food and Drug, Shenzhen Polytechnic University, No. 2190, Liuxian Road, Shenzhen 518055, China; huangll@szpu.edu.cn (L.H.); liumiaoling2021@163.com (M.L.); 2College of Food Science and Technology, Hebei Agricultural University, Baoding 071001, China; bimal@hebau.edu.cn; 3College of Food and Bioengineering, Henan University of Science and Technology, Luoyang 471003, China

**Keywords:** coffee bean, roast degree, variety, terahertz, identification

## Abstract

In this study, terahertz time-domain spectroscopy (THz-TDS) was proposed to identify coffee of three different varieties and three different roasting degrees of one variety. Principal component analysis (PCA) was applied to extract features from frequency-domain spectral data, and the extracted features were used for classification prediction through linear discrimination (LD), support vector machine (SVM), naive Bayes (NB), and k-nearest neighbors (KNN). The classification effect and misclassification of the model were analyzed via confusion matrix. The coffee varieties, namely Catimor, Typica 1, and Typica 2, under the condition of shallow drying were used for comparative tests. The LD classification model combined with PCA had the best effect of dimension reduction classification, while the speed and accuracy reached 20 ms and 100%, respectively. The LD model was found with the highest speed (25 ms) and accuracy (100%) by comparing the classification results of Typica 1 for three different roasting degrees. The coffee bean quality detection method based on THz-TDS combined with a modeling analysis method had a higher accuracy, faster speed, and simpler operation, and it is expected to become an effective detection method in coffee identification.

## 1. Introduction

Coffee is one of the most popular beverages in the world. It has been consumed for hundreds of years and has become an important part of cultural traditions and social life. In the United States, 85% of adults consume an average of 135 mg of caffeine per day, which is equivalent to about 1.5 standard cups of coffee, and coffee is the main source of caffeine intake for adults [1]. In terms of its economic value, coffee is one of the most valuable agricultural products, which has been exported by the third world and developing countries [2]. It is planted and cultivated in the equatorial zone; cherries and beans are the fruits and seeds of a coffee plant [3]. The final quality of green coffee beans depends on the coffee plant cultivar, geography, weather conditions, and infrastructure available. Wet and dry post-harvest processes may affect the final taste of the beverage; however, the final flavor and aroma are mainly influenced by roasting [4]. To obtain a popular beverage, green coffee beans are roasted at temperatures ranging from 176.7 to 232.2 °C with roasting times ranging from 10 to 20 min, producing light to dark roasts [5]. In general, light-roast coffee beans are suitable for pour-over coffee and dark-roast coffee beans are good for espresso. Since the applications of coffee beans with different roast levels are becoming more important, the precise level of roasting has become an indicator of roasted coffee bean applications. 

In the coffee industry, variety identification is generally used for two purposes: for a consistent supply of coffee beans and for the identification of a new variety. The development of single-origin coffee is an important strategy to maintain coffee quality, grade, and a high cupping score [6]. Roasting level identification is used to evaluate whether the original roasting level is maintained from a small batch to a large batch of production. Capillary electrophoresis-mass spectrometry [7], GC/MS [6], high-throughput metabolomics, DNA RFLP fingerprinting [8], hyperspectral imaging [9], Near-Infrared Spectrometer [10], etc., were used to identify coffee varieties. The variety identification was carried out through the separation and identification of a chemical compound in green coffee beans or coffee metabolites for capillary electrophoresis-mass spectrometry and GC/MS. DNA RFLP fingerprinting is a biological detection method, including DNA extraction, PCR amplification, subcloning, and sequencing steps. High-throughput metabolomics is chemical analysis combined with biological information analysis. Hyperspectral imaging and near-infrared spectrometer are spectrum detection methods. The accuracy of chemical analysis combined with biological analysis methods is very high, but the detection process is high-cost, time-consuming, and environmentally unfriendly. For spectrum detection methods, the optical signal is converted into data and then analyzed in combination with mathematical models. The accuracy of spectrum detection methods is not as good as chemical and biological detection methods, but simple pretreatment, time efficiency, and environmental friendliness are advantages of the spectrum detection method. At present, the identification of roasting level uses capillary electrophoresis-mass spectrometry [7], color values, coffee pour-over, etc. The color value method is most commonly used, but it cannot distinguish between light and medium roasting or medium and dark roasting, among different coffee varieties. Sensory evaluation is a very important method to evaluate the quality of coffee. For a given coffee bean, if the taste results of the pour-over method are correlated with the results of an equipment and there are effective associations between sensory evaluation and instrumental data, we can ensure the controllability of coffee bean quality further.

Terahertz (electromagnetic waves at frequencies of 0.1 to 10 THz) detection technology has been utilized in areas such as agricultural breeding [11], medicine [12], biology [13], chemistry [14], and foods [15]. The main advantages of THz spectroscopic detection are that it is nondestructive and environmentally friendly, which enables real-time characterization. Absorption spectra in the THz range provide molecular data in rotational and vibrational modes, which can be used to identify chemical compounds. Furthermore, both frequency-domain and time-domain information are related to the physical structure and chemical composition of a sample [16]. 

THz radiation does not readily penetrate through metals or water like a polar liquid, which makes it one of the most useful applications of THz imaging via the detection of moisture or humidity in materials [17]. However, with the exception of the measured water itself, THz imaging requires the drying of samples to remove interferences from the water signal [18,19]. Some samples contain hydrogen bonds or bioactive macromolecules, which have characteristic absorption peaks in the Terahertz band [20], making it useful for qualitative and quantitative detection [16]. But many samples do not have characteristic absorption peaks, so the analysis of these samples must depend on mathematical models. Suitable mathematical models are established first, and then the reliability of the developed models is confirmed via model training and verification [21,22,23]. Terahertz spectroscopy and SVM (support vector machine) were used to identify foods or biological tissue samples. The results showed that the model provided a fast and accurate method for food detection and origin tracing research in the terahertz band with no obvious absorption peak [24]. In this study, coffee samples from different origins were analyzed, while no specific coffee varieties were given. In fact, there is a huge difference in the quality of coffee beans of different varieties. The most common types of coffee beans on the market are two types: Arabica and Robusta. Arabica coffee beans are suitable for making specialty coffee due to their strong aroma, soft taste, and high acidity. Robusta beans are suitable for making instant coffee and espresso due to their stronger aroma, coarser taste, and greater caffeine content. Typica is the native variety of Arabica, while Catimor is a cross between Arabica and Robusta. It would be more useful if terahertz technology could identify coffee beans from the same origin with different varieties, or from the same variety with different roasting levels. 

The objectives of this study were to use the terahertz spectrum to detect coffee beans and to establish different mathematical models based on the detection data—including linear discriminant (LD), support vector machine (SVM), naive Bayes (NB), and k-nearest neighbor (KNN)—which were then used to obtain the highest accuracy model through model training and comparison so as to accurately identify different varieties of coffee beans with the same and different roasting levels for the same coffee variety. 

## 2. Material and Methods

### 2.1. Experimental Materials and Procedures

The three coffee bean types tested were Catimor (Sun-dried, Puer, Yunnan Province, China), Typica 1 (Sun-dried, Puer, Yunnan Province, China), and Typica 2 (Natural process, Sumatera, Pulau, Indonesia), Appendix A. The three coffee beans were purchased from Box Coffee Co., Ltd., (Shenzhen, China). The FUJIROYAL (FUJIKOUKI Co., Ltd., Kumamoto, Japan) roaster was used to roast the coffee beans for light-, medium-, and dark-roast coffee. The weight of each roasted coffee bean was 200 g, and the initial temperature of roasting was 170 °C. Roast temperature during the roasting process, color value, and sensory flavor description are shown in Table 1. Color values of the coffee bean surface and powdered coffee were tested using a coffee roast analyzer (SYNCFO CRA-01, Jianxin Technology Co., Ltd., Xinbei, Taiwan). Six coffee bean samples, including light-roast Catimor, light-roast Typica 2, light-roast Typica 1, medium-roast Typica 1, dark-roast Typica 1, and green Typica 1 were tested in this experiment. 

Coffee powder was placed in a weighing bottle and put in an oven (DHG-9123A, Shanghai Jinghong Experimental Equipment Co., Ltd., Shanghai, China) at 50 °C for 2 h. Next, it was removed and placed in a brown dryer for cooling. Approximately 150 mg of dried coffee powder was pulverized with a mortar and pestle and then placed in a compression mold (QYL5t, Guangyao Machinery Factory, Jiaxing, China). A pressure of 25 MPa was applied for 3 min to form a 13 mm diameter disk of 100% coffee powder. Then, 125 tablets were obtained for each coffee sample; so, each coffee sample was measured 125 times. The data of 100 repetitions were used to build mathematical models and the data of another 25 were used to verify the models. 

### 2.2. THz-TDS System Introduction

The experimental spectra in this research were obtained using a CCT-1800 Terahertz time-domain spectroscopy (THz-TDS) system (China Communication Technology, Shenzhen, China). The typical transmission TDS set-up is depicted in Figure 1.

The femtosecond laser light was separated by a beam-splitter into two beams: an excitation beam and a probe beam. Terahertz radiation was generated by optical excitation of the THz emitter. The probe beam was optically delayed by a variable delay stage and collimated onto the THz receiver. The emitted terahertz pulse was collected, collimated, and then focused by an off-axis parabolic mirror (OPM1) onto the sample under testing. The terahertz pulses emitted from the sample were then collected and focused using another off-axis parabolic mirror (OPM2) on the surface of the THz receiver [25,26,27]. A waveform comprising the terahertz signal as a function of time was reconstructed by varying the optical time delay.

The spectrometer data acquisition system is designed to detect low frequency and low noise THz signal. The THz emitter is modulated using a pulsed DC bias of 80 V and a modulation frequency of 10 kHz is synchronized with an analogue lock-in amplifier. A current-to-voltage preamplifier is connected to the THz receiver to amplify the current of the THz signal and transmits the signal to the lock-in amplifier to boost the signal-to-noise ratio. Then, a high frequency 24-bit ADC is used to store the THz signal. The spectral range of the TDS system is 0.06–4.5 THz and the spectral resolution is better than 10 GHz. The peak dynamic range is greater than 80 dB. The system is no recalibration needed before or during running.

### 2.3. THz-TDS Detection

The sample was placed in a special test fixture, which was then directly attached to the THz-TDS. The section of the sample to be measured was fixed at the focus-point of the transmitted light, and the cavity was sealed after the sample was covered with a lid. The terahertz emission signal of the test sample was then obtained. During the experiment, the sealed sample chamber was continuously filled with nitrogen gas so that the relative humidity was less than 5% and the temperature was maintained at approximately 25 °C.

The reference signal obtained via passing nitrogen (without sample placement) was *E_ref_*(*ω*) and the sample signal obtained by measuring the sample with a thickness of *d* was *E_sam_*(*ω*). The spectral response function *H*(*ω*) of the sample is expressed by Equation (1) [16,21,28,29].
(1)$$H\left(\omega \right)=\frac{Esam(\omega )}{Eref(\omega )}=\frac{4n}{\left(1+n\right)}e^{-ad/2+j\pi \omega (n-1)d/c}=A\left(\omega \right)e^{-j\varphi (\omega )}$$
where *A*(*ω*) is the amplitude ratio of sample to reference; and *φ(ω)* is the chromatographic difference between the sample and reference. According to the model of optical parameters of the material extraction [30], the refractive index *n*(*ω*) and the absorbance *a*(*ω*) of the sample are determined using Equations (2) and (3), respectively.
(2)$$n\left(\omega \right)=\varphi \left(\omega \right)c/\omega d+1$$
(3)$$a\left(\omega \right)=2ln\left\{\frac{4n(\omega )}{A(\omega ){\left[n\left(\omega \right)+1\right]}^{2}}\right\}$$
where *d* is the thickness of the sample, *ω* is the frequency of the radiation, and *c* is the speed of light in a vacuum.

### 2.4. Dimension Reduction and Visualization

The detection method of coffee quality based on THz-TDS combined with model analysis methods in this study was a new application idea. In order to judge the feasibility of the scheme effectively, it was necessary to reduce the dimension of data for better visualization. The principal component analysis method (PCA) is a simple and effective method of dimension reduction [11]. The data of frequency-domain, time-domain, and absorption were used for the analysis of PCA. First, the response function of each coffee sample was established. Second, LD, SVM, NB, and KNN models in a full-band were established. The accuracy and training time of the model were determined by averaging the results 10 times. The accuracy of the full-band linear discrimination model was the highest. Then, 1–20 principal components were extracted, and the relationship between training time and accuracy was studied. The number of principal components was determined according to the relationship between the number of principal components and the accuracy, as well as the efficiency of the model training time.

### 2.5. Modeling and Model Evaluation

Linear discriminant (LD) is a classical supervised data reduction and classification method to find linear combinations of features that can best distinguish different classes [31,32,33]. In this study, the data reduced by PCA algorithm were input into the LD model, and the average difference between classes was maximized and the variance within classes was minimized. Finally, the classification decision was made via the threshold method. This means that LD clusters data of the same type together, while data of different types are far apart.

Support vector machine (SVM) is a commonly used classifier that essentially searches for the hyperplane with the largest interval in the feature space [34]. In this study, the reduced dimension data of PCA were used as the input of the support vector machine, while the interval maximization learning strategy was used to solve the convex quadratic programming problem. The type parameter of kernel function was selected as quadratic kernel function and the regularization parameter was set to 100, according to the amount of training data.

The naive Bayes (NB) method is a classification method based on Bayes’ theorem and the assumption of conditional independence of features [35,36]. In this study, NB learnt the joint probability distribution based on training data after PCA dimension reduction; then calculated the corresponding posterior probability of each category; and finally selected the category with the largest posterior probability as the prediction result by adjusting the smoothing parameter. According to the size of the dataset and the sparsity of the features, the smoothing parameter was set to 1.0.

K-nearest neighbor (KNN) performs classification by measuring the distance between different feature values [37]. In this study, KNN was calculated using the distance between data after PCA dimension reduction as an indicator of dissimilarity between objects, avoiding the matching problem between objects. Then, the K closest samples were selected as neighbors and classified according to the categories of the neighbors. The algorithm decision distance metric used Euclidean distance and the number of neighbors was selected as 6. Meanwhile, decisions were made based on the dominant categories of multiple objects rather than a single object category.

### 2.6. Data Analysis

The experimental data were analyzed using the statistical software statistic product and service solutions (SPSS, Inc., Chicago, IL, USA) and MATLAB (9.13.0 (R2022b), MathWorks, Natick, MA, USA), and analyses of variance (ANOVAs) were conducted via the ANOVA procedure. All measurements were carried out in triplicate. Mean values were considered significantly different when *p* ≤ 0.05.

## 3. Results and Discussion

### 3.1. THz-TDS Spectral Analysis

Time-domain spectrograms of six coffee bean samples are shown in Figure 2. There were significant differences in the amplitude and phase of the terahertz time-domain spectral signal between the samples and the reference. The Typica 1 coffee bean with different roasting degrees had the same amplitude. The amplitude of Typica 1 green bean was lower, while that of light Catimor and light Typica 2 was the lowest. In terms of phase, compared to the reference, the time-domain signal of Typica 2 coffee bean had a time delay at 4.3 mm and the time-domain signal of other coffee beans had a time delay at 3.9 mm.

The terahertz permeable domain spectrum of the sample was obtained via Fast Fourier Transform (FFT) of the time-domain spectral signal. From Figure 2b, it can be seen that the effective spectral frequency-domain region of the terahertz signal is within 0.0–2.0 THz. The spectrum curves of the tested six coffee beans had the same trend, while the transmitted energy at different frequency points was different. Unlike many complex organisms, coffee beans had no distinct absorption peaks [21,24,38]; in the range of 0.0 to 2.0 THz, there were 436 frequency points. While the high-dimensional Terahertz spectral features bring abundant information; some information with a weak or even irrelevant correlation with sample quality affects the modeling effect. Therefore, principal component analysis (PCA) was first applied in the dimension reduction of terahertz spectral data of different coffee varieties and roasting levels, which were then used for the qualitative identification of these parameters.

### 3.2. Dimension Reduction and Visualization Using PCA

The high-dimensional characteristics of THz-TDS spectrum and the data correction at adjacent frequencies have adverse effects on the model’s performance. To gain insight into the similarity of three coffee bean varieties, the characteristic differences of the same variety under different roasting degrees were statistically analyzed via PCA.

The 3D-PCA is the sample distribution under three principal components obtained via PCA in the frequency-domain of six samples in Figure 3. It can be seen that samples of the same category form clustered clusters, and the clusters of samples of different categories overlap in some areas, indicating that the six samples cannot be completely distinguished using the three principal components. Therefore, PCA dimension reduction analysis was constructed for different types of coffee beans and different roasting degrees of the same coffee beans, which can effectively improve the accuracy of the model and reduce the dimension of the data.

#### 3.2.1. Coffee Bean Varieties

The differences in the THz-TDS spectra of different varieties of coffee bean were the basis for model identification. PCA was used to reduce the sample data to two dimensions and visualization [24]. The distributions of light-roast Typica 1, Typica 2, and Catimor were seen under PC1 and PC2 (Figure 4). Different varieties of coffee beans were obviously aggregated into different clusters when the THz-TDS spectral data were reduced to two dimensions. The data cluster of Catimor and Typica 1 partially crossed and coincided, while the cluster of Typica 2 was distributed independently, which was consistent with the terahertz spectral images of the three varieties. The percentage of data covered by PC1 and PC2 was 96.41%. However, different varieties of coffee beans cannot be completely distinguished by only two principal components. More principal components should be used to input the model for prediction.

#### 3.2.2. Coffee Bean Roasting Degrees

The composition of the same coffee bean gradually changes during roasting. At the initial stage, water evaporation dominates, while carbohydrates and amino acids start to degrade at a later stage of roasting, resulting in changes in color, taste, and nutritional composition. Figure 5 shows the distribution of green bean, light-roast, medium-roast, and dark-roast Typica 1 under PC1 and PC2. The green bean and light-roast Typica 1 were gathered separately in their respective clusters. The distribution of medium-roast and dark-roast Typica 1 overlapped. This might be due to the slow change in terahertz-spectrum-sensitive components in coffee beans from medium-roast to dark-roast. Incomplete and non-uniform roasting might cause confusion in feature distribution. The percent of data explained by PC1 and PC2 was 94.10%. The explanation of different roasting levels also required more principal components.

### 3.3. Results of Modeling and Analysis

#### 3.3.1. Coffee Bean Varieties Prediction Result

Coffee beans with a different roasting degree and different variety were divided into two groups to establish mathematical models. First, three kinds of light roast coffee beans were distinguished via a mathematical model. The response function of each coffee sample was established. Then, LD, SVM, NB, and KNN models in a full-band were established. The accuracy and training time of the model were determined by averaging 10 replicates. The optimal number of principal components was found via PCA. Second, green coffee beans, and light-roast, medium-roast and dark-roast Typica 1 were distinguished via a mathematical model. The working steps were the same as before. Third, six coffee samples were put together to distinguish between them, following the same working procedure as before.

The THz-TDS spectra of Typica 1, Catimor, and Typica 2 (100 per species) with a light roast were used as features for model training and analysis. Independently collected test set response spectra (25 per species) were used to evaluate the classifier. The feature data were input into LD, SVM, NB, and KNN, while the cross-validation accuracy and model training time of the training set were under 5-fold cross-validation (Table 2). All the results shown in the table were the average of 10 independent tests. The LD had the highest cross-validation accuracy with 100%, while the KNN had the lowest time loss of 0.73 s. The time loss of LD and SVM was smaller than that of KNN; however, the accuracy was higher. When classifying independent test sets, LD had both the highest accuracy of 100% and the lowest time loss of 0.034 s. This also showed that there was a linear relationship between the terahertz spectral data of the three kinds of coffee under light roasting.

Through PCA dimension reduction, the correlation between adjacent bands was eliminated and a simpler and faster model was constructed, which was suitable for industrial real-time detection [39]. The test set prediction accuracy of the LD model reached 98%, while the prediction time stabilized at 15 ms under the first five principal components (Figure 6). Such fast detection was able to save 50% of the time compared to the full-band modeling. The first 20 principal components can be selected to pursue high-precision detection. Obviously, the increase in the number of principal components effectively improved the prediction accuracy of the model; at the same time, the prediction time of the model also increased.

#### 3.3.2. Coffee Bean Roasting Degrees Prediction Result

The THz-TDS of green-bean, light-, medium- and dark-roast Typica 1 (100 per species) were used as features for model training and analysis. Independently collected test set response spectra (25 per species) were used to evaluate the classifier. The LD, SVM, NB, and KNN model were used as classifiers to evaluate the roasting degree of Typica 1. The results of 5-fold cross-validation and prediction are shown in Table 3. The LD had the highest cross-validation accuracy of 100%, while the KNN had the lowest time loss of 1.08 s. The time loss of LD and SVM was smaller than that of KNN; however, the accuracy was higher. When classifying independent test sets, LD had both the highest accuracy of 100% and the lowest time loss of 0.054 s.

In order to improve the model robustness and further improve the classification efficiency, feature dimension reduction was performed through PCA. Figure 7 shows the time loss and accuracy of the LD model on the test set when the number of principal components increases. As the number of principal components (the model input features) increased, the LD model prediction time and accuracy gradually increased. After reaching 28 principal components, the prediction accuracy stabilized at 100%, while the prediction time fluctuated around 25 ms. Compared with full-band modeling, the efficiency was increased by 53.7%. In this study, the prediction accuracy reached 100% because there were significant differences between different types of data on a high-dimensional level, and the features of the terahertz spectrum were effectively extracted in modeling analysis and important information was retained. At the same time, the test task was carried out under laboratory conditions, which can effectively reduce the noise and uncertainty of data and provided a great reference value for industrial field application.

#### 3.3.3. Comprehensive Detection of Coffee Beans

All six samples were used for classification. The PCA and LD classifiers were also used to identify the samples; their variety and degree of roasting could not be determined. The classification result of 150 samples (25 per species) under 30 principal components was 95.3%. The confusion matrix of the test set classification is shown in Figure 8. Four samples of medium-roast Typica 1 were predicted to be dark-roast Typica 1; one sample of light-roast Typica 1 was predicted to be medium-roast Typica 1; and two samples of dark-roast Typica 1 were predicted to be medium-roast Typica 1. Parts of the samples were not uniformly roasted; so, the components that were sensitive to the terahertz spectrum changed slightly during the roasting process, causing the model to misidentify. Therefore, different types of coffee beans can still be fully identified.

### 3.4. Discussion

Terahertz detection combined with a mathematical model was used to identify coffee bean variety and roasting degree in this study. According to the results of the data analysis, the PCA method combined with the linear model can used to distinguish six coffee samples. For samples of three different varieties, Catimor and Typica 1 are both from Yunnan, China, while Typica 1 and Typica 2 are native varieties of Arabica. This means that this identification method has the potential to distinguish different varieties of the same origin, as well as the same variety of coffee beans from different origins. Typica 1 coffee beans with different roasting degrees can also be clearly distinguished, which provides method support for distinguishing different roasting degrees of the same variety or different roasting degrees of different varieties. In recent years, there are fewer and fewer areas suitable for the growth of Arabica coffee beans due to climate and other factors. However, the global coffee market has shown an increasing demand for high-quality coffee beans. As mentioned above, researchers are also trying to use different methods to identify the variety, origin, and roasting degree of coffee beans [6,7,8,9,10]. Each method has its own advantages and disadvantages, but, on the whole, a convenient, environmentally friendly, high-accuracy identification method is the most valuable application. Terahertz detection is such a method.

## 4. Conclusions

Although there was no obvious characteristic absorption peak in the terahertz spectrum of coffee beans, there were many data detected using terahertz, including time-domain, frequency-domain, absorption, refractive index, etc. Too much information was not conducive to the establishment of an effective mathematical model because there was a correlation between some data. Dimensionality reduction of data processing can improve the effectiveness of data processing. Meanwhile, too low a dimension can make it difficult to distinguish coffee samples. Among the four models, LD had the highest accuracy, indicating that there was a linear relationship between the terahertz spectra of coffee samples. Therefore, the LD combined with PCA was an effective way to distinguish coffee samples. If more coffee beans, e.g., with a different geographical origin, different treatment method, different harvest year, etc., are to be considered to build the database, multi-category models will be established according to different requirements. Additionally, the model can be trained sufficiently to reduce its error and improve its accuracy. Such strategies have application value for the identification of coffee bean varieties and the commercial classification of coffee beans for evaluating coffee quality during competition activities.

## Figures and Tables

**Figure 1 foods-13-00389-f001:**
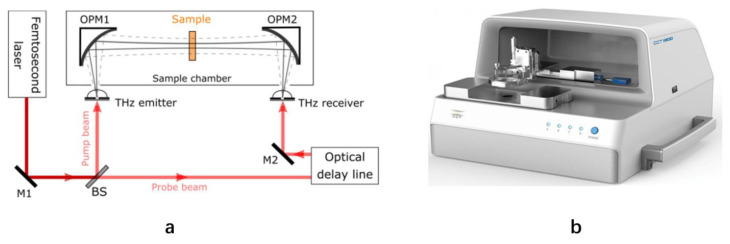
Schematic diagram and picture of transmission-type THz-TDS system. (**a**) Schematic diagram; and (**b**) actual picture.

**Figure 2 foods-13-00389-f002:**
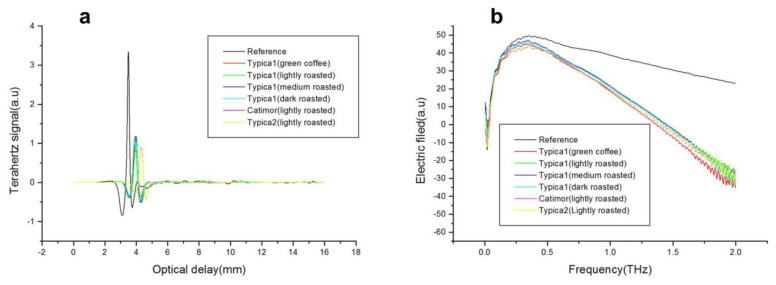
Time-domain spectra (**a**) and frequency-domain spectra (**b**) of six coffee beans with different roasting levels and varieties.

**Figure 3 foods-13-00389-f003:**
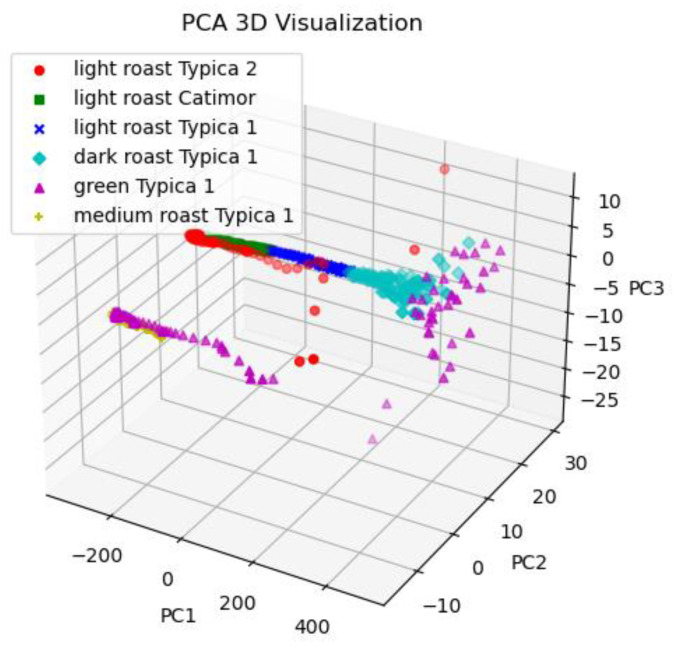
Distribution of six coffee samples under PC1, PC2, and PC3.

**Figure 4 foods-13-00389-f004:**
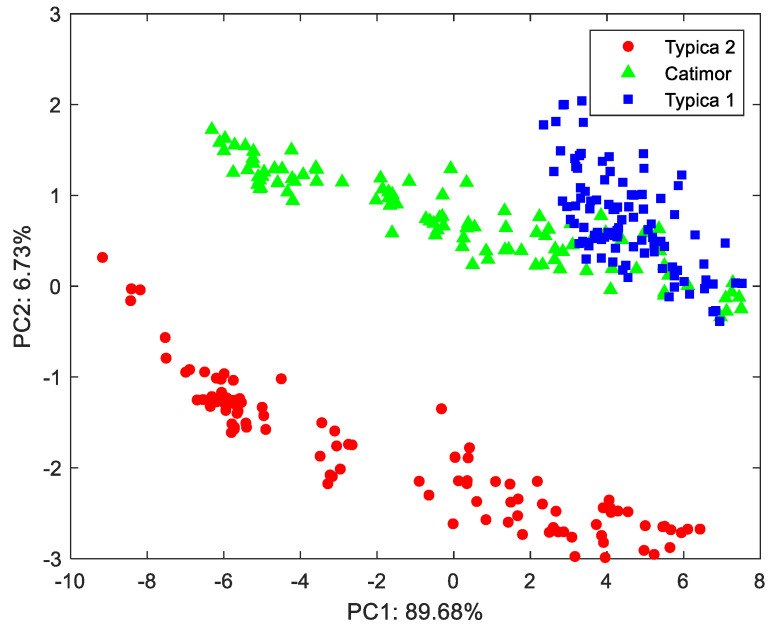
Distribution of light-roast Typica 1, Catimor, and Typica 2 under PC1 and PC2.

**Figure 5 foods-13-00389-f005:**
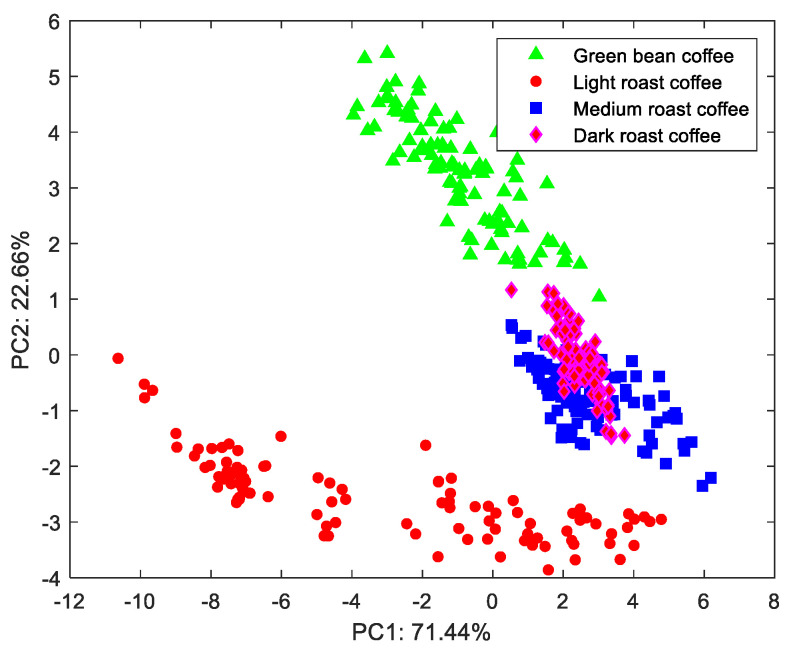
Distribution of green-bean, light-roast, medium-roast, and dark-roast Typica 1 under PC1 and PC2.

**Figure 6 foods-13-00389-f006:**
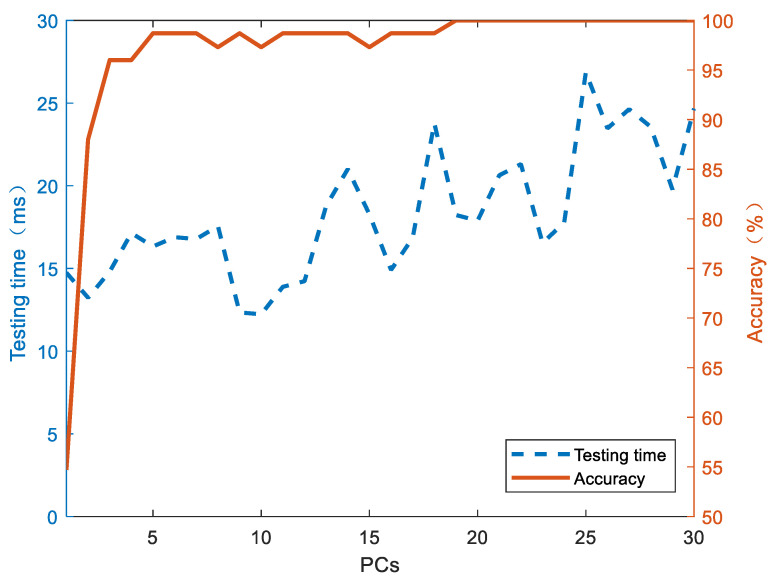
The relationship between testing time and accuracy of the coffee varieties’ classification as the number of PCA principal components increases.

**Figure 7 foods-13-00389-f007:**
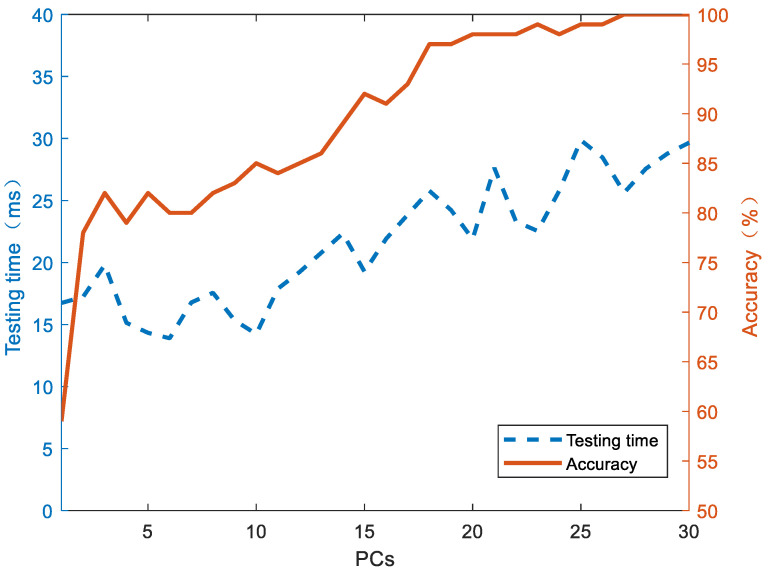
The relationship between the testing time and accuracy of the roasting degree classification as the number of PCA principal components increases.

**Figure 8 foods-13-00389-f008:**
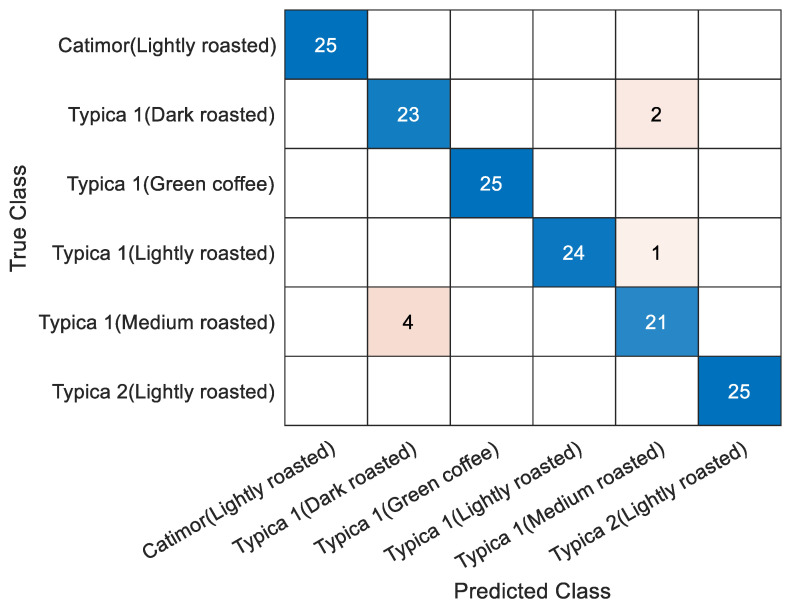
Confusion matrix of the classification of six samples using PCA combined with the LD model.

**Table 1 foods-13-00389-t001:** Roasting conditions of different coffee beans and cupping test result.

	Roasting	Surface Color	Powder Color	First Burst Temperature (°C)	First Burst Time	End Temperature (°C)	End Time	Main Flavor Description
Catimor	Light	68.8 ± 1.6 ^c^	103.0 ± 0.3 ^a^	193	7′50″	200	8′23″	Tropical fruit, fermented
Typica 1	Green	--	--	--	--	--	--	--
Typica 1	Light	84.4 ± 1.9 ^a^	101.8 ± 0.3 ^b^	190	7′40″	200	8′25″	Dried fruit, slightly ripe fruit, woody
Typica 1	Medium	65.8 ± 1.5 ^c^	79.6 ± 0.2 ^d^	190	7′45″	210	9′34″	Citric acid, grape,
Typica 1	Dark	47.5 ± 1.2 ^d^	57.0 ± 0.2 ^e^	190	7′38″	220	10′29″	Roasted nuts, smoky
Typica 2	Light	77.9 ± 1.4 ^b^	92.4 ± 0.2 ^c^	184	7′50″	200	9′36″	Barley tea, vegetative, peanut

“--” means no value due to no roasting of green coffee bean; values in the same column not sharing the same superscript are significantly different (*p* < 0.05).

**Table 2 foods-13-00389-t002:** Cross validation and prediction results of the four models for different varieties of coffee.

Model	Cross-Validation		Prediction	
	Accuracy	Time (s)	Accuracy	Time (s)
LD	100%	1.10	100%	0.034
SVM	100%	1.01	98.7%	0.032
NB	99.7%	20.77	100%	0.402
KNN	100%	0.73	94.7%	0.029

**Table 3 foods-13-00389-t003:** Cross validation and prediction results of the four models for different roasting degrees.

Model	Cross-Validation		Prediction	
	Accuracy	Time (s)	Accuracy	Time (s)
LD	100%	4.31	100%	0.054
SVM	99.0%	3.09	96%	0.052
NB	94.8%	30.89	91%	0.658
KNN	96.5%	1.08	90%	0.029

## Data Availability

Data is contained within the article or Appendix A.

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
