# Peer review of "Terahertz Spectroscopic Identification of Roast Degree and Variety of Coffee Beans"

_foods, 2024, doi:10.3390/foods13030389_

Round 1

Reviewer 1 Report

Comments and Suggestions for Authors

Dear Authors

Your manuscript describes Terahertz Spectroscopic Identification of Roast Degree and Variety of Coffee Beans, which is a new approach in coffee research. However, some paragraphs are not easy to read.

The main purpose of your work is not really clear. You want to detect easily and rather fast the roast degree and the variety of coffee, but your analysis takes several hours for drying and preparing the sample. Especially the detection of the roast degree can be done much faster.

You want to set up a prediction model for the coffee variety. For this, you need much more coffee samples, from different harvesting years etc. Three samples at one roast degree are not enough.

I would suggest revising your manuscript completely and put the focus on the evaluation of the proof of principle of the technique, but for sure not on any prediction model.

Comments on the Quality of English Language

The Quality of English Language is ok, but the red line in paragraphs is sometimes missing, and some have to be rewritten in Materials and Methods:

Page 2, line 48-63: please revise this paragraph completely. Do not mix different topics like roast degree, gene sequencing and sensory.

Page 4 and 5, paragraphs 2.4 and 2.5: please shorten, it is more written like a text book than like a Materials and Methods section. Define in more detail which software you used, which settings in the software, how you prepared your data.

Author Response

Comment: Your manuscript describes Terahertz Spectroscopic Identification of Roast Degree and Variety of Coffee Beans, which is a new approach in coffee research. However, some paragraphs are not easy to read.

Response: Thank you for your careful review and feedback. We have restructured the sentences which are not clear enough to understand (Changes have been highlighted by red texts).

Comment: The main purpose of your work is not really clear. You want to detect easily and rather fast the roast degree and the variety of coffee, but your analysis takes several hours for drying and preparing the sample. Especially the detection of the roast degree can be done much faster.

Response: Thank you for your review and feedback. In this work, terahertz testing was used to identify coffee beans from different origins and coffee beans with different roasting degree. For that coffee beans need to be dried and roasted. Coffee powder was dried in an oven at 50 °C for 2 h in order to remove the effects of water present in coffee powder. To clarify the procedure and method, few descriptions in Abstract and Introduction sections have been revised (red texts).

Comment: You want to set up a prediction model for the coffee variety. For this, you need much more coffee samples, from different harvesting years etc. Three samples at one roast degree are not enough.

Response: Thank you for your suggestion. Three coffee beans with different variety and one of them (Typica 1) with different roasting degree were tested. The more the number of samples, the more complex the model built will be. There is no obvious peak of coffee samples. The main purpose of this study is to distinguish the coffee beans of different varieties and different roasting degrees through the establishment of mathematical models. The reviewer's opinion is very good and we will consider your suggestion of considering more samples to develop the model in our upcoming work to analyze the difference between harvesting years of coffee beans. However, author’s team has already carried out terahertz detection of different years of dried tangerine peel.

Comment: I would suggest revising your manuscript completely and put the focus on the evaluation of the proof of principle of the technique, but for sure not on any prediction model.

Response: Thank you for your suggestion and feedback. Authors check this manuscript completely and focus on the evaluation of the proof of principle of the technique. At present, coffee identification is mostly based on sensory evaluation. So, it is necessary to establish an objective evaluation method. The goal of this study is to try to establish an objective method based on detection results and mathematical analysis. To address this, some descriptions in Material and Methods have been revised.

Comments on the Quality of English Language

The Quality of English Language is ok, but the red line in paragraphs is sometimes missing, and some have to be rewritten in Materials and Methods.

Response: Thank you for your positive comments about our English language. We have re-written the ambiguous part of Materials and Methods and others as suggested.

Comment: Page 2, line 48-63: please revise this paragraph completely. Do not mix different topics like roast degree, gene sequencing and sensory.

Response: Thank you for your review in details. Following your suggestion, the stated paragraph has been revised.

Comment: Page 4 and 5, paragraphs 2.4 and 2.5: please shorten, it is more written like a text book than like a Materials and Methods section. Define in more detail which software you used, which settings in the software, how you prepared your data.

Response: Thanks for your comments. According to your suggestion, authors revised paragraphs 2.4 and 2.5 to make them more concise and focus on the parameter setting of the method in application.

Reviewer 2 Report

Comments and Suggestions for Authors

Overall Comments

The authors presented an article entitled “Terahertz Spectroscopic Identification of Roast Degree and Variety of Coffee Beans” in which classification models are employed in order to differentiate coffee types, roasting and other attributes.  The English grammar is good. The scientific design was hard to evaluate due to the lack of clarity in the presented information. The idea of the article is good and the use of an incommon method such as Terahertz Spectroscopy is interesting. However, the manuscript lacks in clarity in many points of the work. The employed dataset is not clearly stated in the methodology section. The division within training and validation groups is not clear. The data processing is not clearly stated. The specters parameters employed are also elusive in the text. The 100% accuracy could indicate overfitting, however, there is no discussion or attempt in improving the model. Validation parameters for the classification are missing. The classification matrix could be presented for every prediction, but is only shown in the section 3.3.3.

Specidic comments

L 134 -153 It is important that the authors, specify the range of the resulting THz-TDS specter and the number of variables; as is, there is no way to know if the dimensionality reduction method is even necessary.

Another unclear information is about the samples employed in the classification model. Did the authors used the 6 samples presented in Table 1? This information must be clearly stated in the text

L200 “The THz-TDS spectroscopy data contains all the necessary information.” What did the authors mean by necessary information? How was that ascertained? Necessary for what exactly? The text is ambiguous

Figure 3. The PCA seemed to have withdrawn much of the explained variance within the first components, which indicates a good correlation of the data and the data is clearly displaying cluster, another good sign for classification. There seemed to have been used a great sample size judging by the number of points, however, how many where used? They were all from the same source or are those replicates?

Without clearly understanding the employed samples it is hard to evaluate the findings and even the figures. This information must be presented in the methodology.

Figure 4. The overlap in the Dark roast and Medium roast maybe could be elucidated using PC2 and PC3, for this particular duo; thus, classification could improve

L 266 – 285 The validation results also, are hard to evaluate, without knowing the size and proper nature of the specter dataset employed. There seems to be an overfitting in LD and KNN for the coffee variety prediction. 100% accuracy with cross-validation are rarely representative of real applicable models  

Author Response

Comment: The authors presented an article entitled “Terahertz Spectroscopic Identification of Roast Degree and Variety of Coffee Beans” in which classification models are employed in order to differentiate coffee types, roasting and other attributes.  The English grammar is good. The scientific design was hard to evaluate due to the lack of clarity in the presented information. The idea of the article is good and the use of an incommon method such as Terahertz Spectroscopy is interesting. However, the manuscript lacks in clarity in many points of the work. The employed dataset is not clearly stated in the methodology section. The division within training and validation groups is not clear. The data processing is not clearly stated. The specters parameters employed are also elusive in the text. The 100% accuracy could indicate overfitting, however, there is no discussion or attempt in improving the model. Validation parameters for the classification are missing. The classification matrix could be presented for every prediction, but is only shown in the section 3.3.3. 

Response: Thank you for your review with affirmative comments. We have checked the whole manuscript and many descriptions have been revised to address the points raised, including methodology section. In this study, the prediction accuracy reaches 100%, because there are significant differences between different types of data in high-dimensional level, and the features of terahertz spectrum are effectively extracted in modeling analysis, and important information is retained. At the same time, the test task is carried out under laboratory conditions, which can effectively reduce the noise and uncertainty of data, and provide great reference value for industrial field application. We added the relevant description in the revised manuscript.

The prediction accuracy of the model is the decisive index of model goodness. The model with the best performance is selected through the prediction accuracy, while the classification matrix (confusion matrix) is only used to analyze the identification results of the model for each type of sample. For the model with low prediction accuracy, the classification matrix is not necessary.

Specific comments

Comment: L 134 -153 It is important that the authors, specify the range of the resulting THz-TDS specter and the number of variables; as is, there is no way to know if the dimensionality reduction method is even necessary.

Response: Thank you for your feedback with your viewpoint. Compared with the detection task, THz-TDS spectrum has higher dimension, and its adjacent frequency bands have high correlation. Considering the time and calculation cost of high latitude, it is necessary to use dimension reduction means. Meanwhile, specifying exactly the range of THz-TDS spectrum and the number of variables can improve the repeatability and comparability of the research.

Comment: Another unclear information is about the samples employed in the classification model. Did the authors used the 6 samples presented in Table 1? This information must be clearly stated in the text

Response: Thank you for your question for clarity. Yes, six samples were used in classification model. This sentence has been added in section 3.3.3.

Comment: L200 “The THz-TDS spectroscopy data contains all the necessary information.” What did the authors mean by necessary information? How was that ascertained? Necessary for what exactly? The text is ambiguous 

Response: Thank you for your multiple questions to make the statement clear. We wanted to say that the terahertz assay had obtained all the data necessary for mathematical analysis but this sentence was redundant here. Therefore, this statement has been removed.

Comment: Figure 3. The PCA seemed to have withdrawn much of the explained variance within the first components, which indicates a good correlation of the data and the data is clearly displaying cluster, another good sign for classification. There seemed to have been used a great sample size judging by the number of points, however, how many where used? They were all from the same source or are those replicates?

Without clearly understanding the employed samples it is hard to evaluate the findings and even the figures. This information must be presented in the methodology. 

Response: Thank you for these questions to make it clear. Altogether 125 tablets for each coffee sample were tested to obtain the data for model building and model verification. In fact, the number of tablets were higher than 125 because they were easy to break; however, we assured the measurement of 125 tests. Among them the data of 100 samples were used for model building, while remaining 25 data were used for model verification. This information has been added in section 2.1.

Comment: Figure 4. The overlap in the Dark roast and Medium roast maybe could be elucidated using PC2 and PC3, for this particular duo; thus, classification could improve

Response: Thank you for your suggestion. Perhaps PC3 can be considered to distinguish between dark roast coffee and medium roast coffee. However, this was not done in this work and we will do it in our next work.  

Comment: L 266 – 285 The validation results also, are hard to evaluate, without knowing the size and proper nature of the specter dataset employed. There seems to be an overfitting in LD and KNN for the coffee variety prediction. 100% accuracy with cross-validation are rarely representative of real applicable models.   

Response: Thank you for your viewpoints with good idea. The size and proper nature of the specter dataset employed are very important. So, the model of the terahertz device and the weight and size of the flake coffee sample were shown in manuscript.

In this study, the prediction accuracy reaches 100%, because there are significant differences between different types of data in high-dimensional level, and the features of terahertz spectrum are effectively extracted in modeling analysis, and important information is retained. At the same time, the test task is carried out under laboratory conditions, which can effectively reduce the noise and uncertainty of data, and provide great reference value for industrial field application. We added the relevant description in the paper.

Reviewer 3 Report

Comments and Suggestions for Authors

I made the review with some delay, which is due to a rather interesting and innovative test used to analyze the degree of coffee roasting and adapting its taste to consumer habits. The authors planned quite a thoughtful experiment. used an interesting analytical method which, thanks to machine statistics, allowed them to develop models for recognizing the degree of coffee roasting. the whole work was prepared quite well. I wonder whether the results obtained - the developed method will also allow the evaluation of Arabica or Robusta coffees. and also to what extent the complicated method can be used by producers or coffee roasters.

Author Response

Commets: I made the review with some delay, which is due to a rather interesting and innovative test used to analyze the degree of coffee roasting and adapting its taste to consumer habits. The authors planned quite a thoughtful experiment. used an interesting analytical method which, thanks to machine statistics, allowed them to develop models for recognizing the degree of coffee roasting. the whole work was prepared quite well. I wonder whether the results obtained - the developed method will also allow the evaluation of Arabica or Robusta coffees. and also to what extent the complicated method can be used by producers or coffee roasters.

Response: Thank you for your viewpoints with positive comments. In this work, Typica is the native variety of Arabica, while Catimor is a cross between the Arabica and Robusta. They can be distinguished by THz spectroscopy combined with mathematical models. So, the developed method will also allow the evaluation of Arabica or Robusta coffees. Authors interviewed many experts in the coffee industry while conducting this study; with the increasing scale of industrial production, the precision of coffee bean quality is increasingly required. They are very excited to have easy to operate, low cost, environmentally friendly testing methods. Terahertz detection could be one of such methods.  

Reviewer 4 Report

Comments and Suggestions for Authors

The authors have presented a promising study about a proposed sytems to classify cofee beans according to variety and roast degree. Different mathematical methos have been proposed.

A clear relation of the classification procedures and the product could be included.

- Line 53: different format.

- The practical objective of the study needs to be explained. Is there a problem identifying different varieties of coffee (and roast degree)?

- Introduction: a review of other techniques used for the Identification of roast degree and variety of coffee beans could be added.

- A description of the different coffee beans varieties could be added.

- “2. Material and methods”: the experimental design need to be clarified (factors, measured variables, repetitions).

- It is necessary to include some photos of the different coffee bean varieties and roast levels.

- Lines 154-198: the description of the analysis methods could have been reviewed in the introduction.

- The relation of the measured variables to the principal component in the PCA should be shown.

- The mathematical models addressed in the results could be referenced to the different coffee bean characteristics.

- A discussion of the results needs to be included.

- The conclusions are poor. a reflexion about the importance of the proposed classification methodology (using the different data analysis) should be included.

Author Response

Comment: The authors have presented a promising study about a proposed system to classify coffee beans according to variety and roast degree. Different mathematical methods have been proposed.

A clear relation of the classification procedures and the product could be included.

- Line 53: different format.

Response: Thank you. The format of whole manuscript was checked and revised.

Comment: The practical objective of the study needs to be explained. Is there a problem identifying different varieties of coffee (and roast degree)?

Response: Thank you for your query. For different roast degree, the main method is to detect the color value. There are several common ways to identify a variety. The first is to know which tree the coffee comes from. The second is that some coffee beans have a special shape. The most rigorous method is to use gene sequencing, but the cost is high. With the increasing industrial processing of coffee beans, if a more practical detection method for the identification of coffee varieties was developed. We believe that it will be of great application value to coffee industry.

Comment: Introduction: a review of other techniques used for the Identification of roast degree and variety of coffee beans could be added.

Response: Thank you for your suggestion. Some techniques used for identification of roast degree and variety of coffee beans have been added in the second paragraph.

Comment: A description of the different coffee beans varieties could be added.

Response: Thank you for your suggestion. The description of the different coffee beans varieties has been added in the fourth paragraph.

Comment: “2. Material and methods”: the experimental design need to be clarified (factors, measured variables, repetitions).

Response: Thank you for your comment. The experimental design has been checked and factors, measured variables and repetitions have been added and clarified (Changes have been highlighted by red texts).

Comment: It is necessary to include some photos of the different coffee bean varieties and roast levels.

Response: Thank you for your suggestion. Some coffee pictures with different varieties and roast levels were shown in a document for supplementary materials.

Comment: Lines 154-198: the description of the analysis methods could have been reviewed in the introduction.

Response: Thank you for your suggestion. The description of the analysis methods (LD, SVM, NB and KNN) has been reviewed in the introduction section.

Comment: The relation of the measured variables to the principal component in the PCA should be shown.

Response: We have rewritten paragraph 2.4. THz-TDS spectrum variables are transformed into a new set of principal components by PCA, which are linear combinations of variables and are used as inputs of prediction models. The visualization of principal components is shown in Figures 3 and 4 in the results section.

Comment: The mathematical models addressed in the results could be referenced to the different coffee bean characteristics.

Response: Thank you for your suggestion. Next, our team will carry out research on the detection of the same variety of coffee beans in different harvesting years to analyze the differences between coffee beans more comprehensively.

Comment: A discussion of the results needs to be included.

Response: The results have been checked and some discussion was added in this part. The new added discussion content was highlighted in red.

Comment: The conclusions are poor. a reflexion about the importance of the proposed classification methodology (using the different data analysis) should be included.

Response: Thank you for your feedback. A reflexion about the importance of the proposed classification methodology was added in the conclusion. The new added content was highlighted in red.

Round 2

Reviewer 1 Report

Comments and Suggestions for Authors

Dear Authors

Thank you for revising your manuscript.

Now the research is presented comprehensively.

Congratulations to your very interesting work.

Reviewer 4 Report

Comments and Suggestions for Authors

The authors have addressed some of the reviewer comments. However, some crucial considerations still need to be addressed and included (or modified).

Line numbers should be added.

Figures should be included in the test.

Figure 1 should be modified by an actual photo of the device (or added).

Detailed information about the THz-TDS system data acquisition and the signal processing should be added.

First paragraph from section 2.2. “Six coffee bean samples, including light roast Catimor, light roast Typica 2, light roast Typica 1, medium roast Typica 1, dark roast Typica 1 and green Typica 1 were tested in this experiment”. According to this sentence it seems that the experimental design was built with two factors but three varieties and only two roasting degrees. However, in the abstract it has been added “Three different varieties and three different roasting degrees” (that would be 9 coffee beans samples not 6). Besides, in Figure 4 there are 4 roasting degrees. This important confusion needs to be clarified.

Information about the PCA methodology has been included in M & M section. However, this information should be summarised.

A Principal Component Analysis (PCA) plot showing the Terahertz time-domain spectroscopy variables should be added (with vectors indicating the direction and strength of each variable).

Figure 3 and 4 should be modified by an only Figure (PCA plot).  It is crucial to develop a Principal Component Analysis to separate the different groups (variety & roasting degree). An only PCA plot should be found separating the different groups (variety & roasting degree).

A Discussion section should be added.

The conclusions are still confusing and should be improved.
